# What Type of Person Should I Be? About the Appeal to Virtues in Public Health Interventions

**DOI:** 10.3390/vaccines11040767

**Published:** 2023-03-30

**Authors:** Pietro Refolo, Dario Sacchini, Costanza Raimondi, Giovanna Elisa Calabrò, Antonio Gioacchino Spagnolo

**Affiliations:** 1Section of Bioethics and Medical Humanities, Department of Healthcare Surveillance and Bioethics, Università Cattolica del Sacro Cuore, Largo Francesco Vito 1, 00168 Rome, Italy; pietro.refolo@unicatt.it (P.R.); costanza.raimondi1@unicatt.it (C.R.);; 2Research Center for Clinical Bioethics and Medical Humanities, Università Cattolica del Sacro Cuore, Largo Francesco Vito 1, 00168 Rome, Italy; 3Section of Hygiene, Department of Life Sciences and Public Health, Università Cattolica del Sacro Cuore, Largo Francesco Vito 1, 00168 Rome, Italy; 4VIHTALI (Value in Health Technology and Academy for Leadership & Innovation), Spin-Off of Università Cattolica del Sacro Cuore, 00168 Rome, Italy

**Keywords:** public health, public health ethics, deontologism, consequentialism, utilitarianism, virtues ethics, vaccination

## Abstract

In line with how ethics has developed for the last three centuries, public health ethics has been widely dominated by a deontological as well as a utilitarian approach. The latter is a version of consequentialism, which states that maximizing utility is the primary goal of the majority of individuals or group action, while, on the other hand, virtue ethics, or at least the appeal to virtues, has been largely marginalized. The aim of this article is twofold. Firstly, we aim to highlight the political and ethical nature of public health interventions, often interpreted and presented as mere scientific enterprises. Secondly, we try to highlight the need to integrate or at least recognize the value of appeal to virtues in public health measures. The analysis will reference the Italian COVID-19 vaccination program as a case study. Initially, we will explore the political and ethical nature of any public health measure, using the implementation of the COVID-19 vaccination program in Italy as an example. Subsequently, we will illustrate the deontological approach to ethics, the utilitarian one, and the virtues one, focusing on the dynamic of the agent’s perspective. Lastly, we will briefly analyze both the Italian COVID-19 vaccination program and the communication campaign that promoted it.

## 1. Introduction

For the past three centuries, virtue ethics, or at least the appeal to virtues, has been widely marginalized in the field of ethics in favor of deontological and consequentialist approaches. The latter essentially state that what matters are the principles, rules, and obligations. Deontological ethics, or deontologism, argues that duties and rules are the basis of morality, while consequentialism holds that the morality of an action is connected to its effects.

One of the main objections against these two approaches is that they reduce morality to the mere fulfilment of a set of duties, obligations, or rules. In addition, they transform the agent into a sort of “moral machine” that “has been set to function in a certain defined way, or an animal that has been trained to do things in a way wanted by the trainer” [1]. In this approach, the “virtue” that lies behind the duties or rules is not looked at. In fact, both the deontological and the consequentialist approaches answer the question “what should I do?” but are not capable nor are they interested in answering the question “how should I live my life?” or “what type of person should I be?” In short, the character of the person who performs the act is ultimately irrelevant in both deontological and consequentialist ethics.

Public health has been defined in several ways [2,3,4] over the past few decades, reflecting its diverse contexts, activities, and interventions. Differently from clinical practice, characterized by a personal physician–patient relationship, public health practice is essentially characterized by global attention to entire populations and, therefore, by an emphasis on collective health conditions, prevention, and social, economic, and demographic determinants of health and disease [4]. Vaccination plans represent the most used tool for primary prevention of disease and, indeed, they are one of the most cost-effective public health measures available.

Public health ethics is the ethics behind public health interventions: it essentially concerns the moral justification of policies, programs, and laws to protect and promote public health [5]. Aligned with the concept of public health, there is no agreement upon the moral concepts and methods for public health ethics. However, in line with the development of ethics as seen in the last three centuries, public health ethics has been largely dominated by the deontological and utilitarian approaches. The latter is a version of consequentialism, which states that maximizing utility is the primary goal of the majority of individuals or group action. Some suggest that public health ethics is in itself utilitarian [6]. As it is the case of ethics at a broader scale, no space is given to the perspective of the agent.

In contrast, public health communication campaigns—i.e., the communication strategies and initiatives that specialists in the field of health promotion engage in at a macro scale to help large groups of people with imminent health threats and to adopt behaviors that promote good health—typically operate by appealing en masse to the virtues of the agents.

The aim of this article is twofold. Firstly, we aim to highlight the political and ethical nature of public health measures, often interpreted and presented as mere scientific enterprises. Before tackling public health ethics issues, we have to recognize the role of ethics in public health interventions. Secondly, we want to highlight the need to integrate or at least recognize the value of the appeal to virtues in public health measures. Despite the irrelevance of the perspective of the person, as is found in both deontologism and utilitarianism, the appeal to virtues of the agent ends up being crucial to the success of public health measures. The following analysis will focus on the Italian COVID-19 vaccination campaign as an example.

## 2. Materials and Methods

One of the authors (PR) identified relevant papers already known to him in order to illustrate the ethical nature of public health interventions. Notably, two articles [7,8] were helpful in structuring the discussion and they represented the basis for analysis of the COVID-19 vaccination program’s implementation in Italy. The authors critically analyzed the collected material and the results were used to identify the “lines” of the issues to discuss. An early draft analysis was prepared by PR and was used by all authors for further discussion. The paragraph/argumentation concerning the ethical nature of public health interventions is the output of such discussion and subsequent revisions by the authors.

Ethical analysis about the deontological, utilitarian, and virtues approach to ethics reflects the authors’ experience in the field of bioethics.

As for vaccine communication campaign, the authors conducted an online search using selected keywords. The search was restricted to institutional videos starting from 2020. Seven institutional videos published by the Presidency of the Council of Ministers (17 January 2021; 21 June 2021; 14 November 2021; 28 December 2021; 4 July 2022; 28 July 2022; and 13 December 2022) [9,10] were retrieved. The authors critically analyzed the collected material, and the results were used to perform the analysis. An early draft analysis was prepared by PR and this became the focus/starting point for further discussion by all authors. The paragraph/argumentation concerning vaccine communication campaign is the output of such discussion and subsequent revisions by the authors.

Finally, the authors combined together the different components of the analysis. Looking at the whole picture, they reorganized the analysis into five thematic areas: ethics behind public health interventions; deontological approach; utilitarian approach; virtue ethics; and Italian COVID-19 vaccination program.

## 3. Results

We report below the results of the five thematic areas.

### 3.1. Ethics behind Public Health Interventions

Many countries developed plans or priority setting documents for the COVID-19 vaccination. In Italy, the COVID-19 vaccination program was essentially based on the “Strategic Plan for anti-SARS-CoV-2/COVID-19 vaccination” [11]. This document was published on 2 January 2021 and was drafted by a number of scientific and policy institutions, including the Ministry of Health, the Extraordinary Commissioner for the COVID-19 Emergency, the Istituto Superiore di Sanità (ISS), the Italian National Agency for Regional Healthcare Services (AGENAS), and the Italian Medicines Agency (AIFA). The plan was subject to adjustments based on new information coming in as a result of continuing scientific research. As such, the Health Ministry released an updated version of the priority groups’ list on 8 February 2021 [12]. Two further updates with minor changes were, respectively, published on 9 April 2021 and on 13 March 2021 [13] by the Extraordinary Commissioner for the COVID-19 Emergency. Our analysis refers to the first two documents, since the remaining two documents contain only technical updates.

Both documents state that the vaccination plan had only one goal, which was common to many countries: to reduce mortality and morbidity for the population. The two documents mention a long list of ethical principles, including human well-being, equal respect, global equity, national equity, reciprocity, and legitimacy, which formed the basis for justifying the proposed priority groups’ list. However, the aforementioned principles are the sole explicit “ethical considerations” mentioned in either text. This can be seen as clear proof—as also pointed out by other authors [8,14,15]—that the Italian program for anti-COVID-19 vaccination contains very limited ethical reasoning and, more broadly speaking, that the issue of ethics behind this public health intervention is never properly addressed.

This issue was not limited to Italy but can be found in other countries as well, such as Spain or Sweden [15]. What we are emphasizing here is that—independently from the ethical approach which inspires a certain public health intervention—vaccination plans are often interpreted and presented as mere scientific decisions. A further tangible sign of this structural issue is the limited public debate that arose after the drafting of national pandemic COVID-19 vaccination plan documents, as well as their high level of technical content.

In this way, vaccination programs seem to be more technical than ethical and more of an organizational matter than a social one; they seem to belong more within the realm of “scientific facts” than within that of “values”, and a sort of “positivistic approach” to the questions seems to be still in use [16].

In contrast, in this context, one should recognize that the role of science is merely instrumental to human purposes. Science can inform us about the effects (positive or negative) of various vaccine strategies, but what we want to achieve depends on which values we think matter the most [17]. On this basis, any plan to use pandemic vaccines should provide robust ethical considerations and ethics behind public health interventions should be made clearly explicit.

### 3.2. Deontological Approach

The term “deontology” derives from the Greek root “deon”, or duty, and “logos”, or science. Hence, deontology can be defined as the “science of duty”. The key philosophers who proposed and supported this approach are Immanuel Kant, William D. Ross, John Rawls, and Robert Nozick, although deontologism is mainly associated with the name of Kant. Kant’s deontological philosophy stems from his belief that humans possess the ability to reason and understand “universal” moral laws that they can apply in all situations. Therefore, the theory of deontology essentially states that what determines what one ought to do is that which is universalizable.

Three fundamental characteristics can be identified to describe this model: firstly, it claims not to rely on any theory of human nature; secondly, there is the priority of the right over the good; and, thirdly, morality lies in principles that are universalizable.

One important consequence is that, in the deontological approach, there is no clear relationship between one person and another; rather, the relationship lies between a person and the duties or rules that one must follow. This could result in morality being perceived as “impersonal”. As noted by Ntibagirirwa [1], “Kantians dehumanize person by giving priority to principles and duties while appearing to ignore the fact that human beings determine these principles and obligations of duty”.

In other words, one of the main risks associated with deontologism is lack of motivation. Why should one improve? In addition, if one should in fact improve, in what way? What type of person should I be? For a deontological approach to ethics, the character of the moral agent is irrelevant. No space is given to the perspective of the person. On the other hand, impartiality is one of the strengths of this approach; that is why it is largely used and appreciated in pluralistic societies.

### 3.3. Utilitarian Approach

Consequentialism is a theory that says whether something is good or bad depending on its outcomes. The most famous and used version of this theory, especially within the healthcare context, is utilitarianism. The key philosophers who proposed and supported this approach are Jeremy Bentham and John Stuart Mill.

Although several varieties of utilitarianism allow for different characterizations, the basic idea behind them all is to maximize utility, which is often defined in terms of well-being or related concepts. In other words, utilitarianism states that maximizing utility is the main goal of the majority of individuals or group action. An action (or type of action) is right if it tends to promote utility and wrong if it does not—not just for the performer of the action, but also for everyone else affected by it (the greatest utility for the greatest number). In fact, “aggregationism” is one of the most important hallmarks of this theory. More specifically, aggregationism is the view according to which “the value of the world is the sum of the values of its parts, where these parts are local phenomena such as experiences, lives, or societies” [17]. For example, utilitarianism claims that improving five lives by a certain amount is better than improving one life by the same amount, five times better.

One important consequence is that, since utilitarianism is a maximization of ethics, it cannot conceive the good of all. That is, there is always the problem of the “some”. Correspondingly, as noted by Ntibagirirwa [1], “the impartiality which is supposed to be the aim of utilitarianism becomes problematic since some people are left out and/or used as a means to achieve utility”. Secondly, the greatest expression of utility suggested by the utilitarian approach is an aggregation of individual goods, and this could also leave room for individualism, which we aim to avoid. In any case, the utilitarian approach does not explicitly state how we can bring the individual to self-sacrifice for others. In this sense, the character of the moral agent is not relevant and no space is given to the perspective of individuals.

### 3.4. Virtue Ethics

The word virtue comes from the Latin root “vir”, for man. Thus, virtue means “(hu)maness” and can also mean “excellence” in more general terms. In virtue ethics, the concept of moral virtue is central. The major proponents are the classical philosophers Socrates, Plato, Aristotle, Augustine of Hippo, Thomas Aquinas, and, in the contemporary age, the moralists Gertrude E.M. Anscombe and Alasdair MacIntyre.

While deontologism and utilitarianism start from the question of “what should be done?”, virtue ethics addresses the question “what should one be?” or “what type of person should I be?”, and the answer provided by this approach is “virtuously!”. Virtue is primarily a disposition of the agent, and, in this sense, virtue ethics is an agent-centered ethics, i.e., a model of ethics that primarily values human beings for their own sake.

Virtue ethics is not interested in the act for its own sake or utility for its own sake but, rather, it is interested in acts and utility to the extent they bring the individual to self-realization and self-perfection, not only for him/herself, but also for the community which provides him/her with the necessary conditions for achieving such self-realization and self-perfection. Therefore, virtues are lived within the community and behaviors are virtuous in as much as they benefit the individual for himself/herself and his/her community. As noted by Ntibagirirwa [1], “the lesson of virtue ethics is that we cannot be moral for the sake of duty or for the sake of utility, we are moral primarily because that is the way we are made to be as social beings”. To be clearer, virtue ethics argues that morality should not be thought of in terms of impersonal duty or impartial maximization of well-being but in terms of a personal flourishing connected to others in special relationships.

However, there has been a historical variation in what was seen as virtuous throughout time; nowadays, the content of the virtue has greatly changed. Therefore, a sort of “moral relativism” or lack of impartiality is considered one of the weaknesses of this approach.

### 3.5. Italian COVID-19 Vaccination Program

The ethical analysis of the Italian Strategic Plan for anti-SARS-CoV-2/COVID-19 vaccination and its updates is fairly easy to carry out, since we have only one reference point, represented by the set of principles mentioned in the third paragraph of this work.

What can be added here is that, even though not explicitly said, the six principles listed are those that are considered to be relevant to vaccination distribution by “WHO SAGE values framework for the allocation and prioritization of COVID-19 vaccination” [18]. The purpose of this document is to offer guidance for the distribution of COVID-19 vaccines among countries, both at a global and a national level, prioritizing certain groups within each country due to limited supply. Specifically, the WHO SAGE framework articulates the overall goal of COVID-19 vaccine deployment (i.e., “to contribute significantly to the equitable protection and promotion of human well-being among all people of the world”) and provides “six core principles that should guide distribution and twelve objectives that further specify the six principles” [18].

The Strategic Plan was accompanied by a vaccine communication campaign. This campaign consisted of institutional videos, radio messages, and articles. Our analysis focused on institutional videos. The key message of the retrieved videos (see Section 2 “materials and methods”) was, of course, the explicit invitation to the entire population to receive the vaccine. Two of these videos (14 November 2021 and 28 July 2022) can be considered as merely informative in the sense that they provided clear and simple information about the vaccination. The remaining five videos can be considered as emotional in their nature in the sense that they tried to reach the citizens at an emotional level. What is noteworthy is that, even though not explicitly said, these videos in truth seem to appeal to the citizen’s virtue, specifically, compassion, courage, and generosity. A similar situation occurred in Italy with the COVID-19 governmental restrictions: on the one hand, a large number of norms and rules of public health were established through governmental decrees (for example, there were even restrictions placed on the maximum number of people who could gather together for Christmas celebrations); on the other hand, a massive communication campaign was set in motion, inviting citizens to be responsible, prudent, and generous.

This phenomenon, however, was not exclusive to Italy. As noted by Moulin-Stożek et al. [19], different actors appeared to have appealed to citizen virtues during the pandemic. For example, the Queen of England spoke of “instinctive compassion”; the World Health Organization (WHO) included two virtues (solidarity and cooperation) in its documents; and the United Nations Educational, Scientific and Cultural Organization (UNESCO) referred to mutual caring, solidarity, and empathy.

Therefore, we can see that appealing to virtues does not seem to be outside the realm of public health interventions.

## 4. Discussion

The aim of this article is not to criticize the deontological approach to ethics or utilitarianism, nor is it to propose a new hierarchy of values or to say what good and bad ethics are. More simply, the aim is, on the one hand, to expose the nature of public health interventions and, on the other hand, to show the irrelevance of the citizen’s perspective for deontologism and utilitarianism (which is a well-known argument in the field of philosophy) in the first place and to show proof, through practical examples, of attempted appeals to virtues in public health interventions in the second place. Similar appeals should have no place in policies inspired by deontologism or by utilitarianism. Therefore, that presence could be indicative of some lack.

In our understanding, what is lacking, in short, is a strong motivation for action. Kantian or utilitarian obligation run the risk of remaining too abstract to provide helpful guidance in everyday life, while virtue ethics focuses on human life. As noted by MacIntyre [20], appeal to virtues can be an efficient remedy against “personal alienation”.

Not by chance, there was a renewed interest in virtue ethics in public and international policy in recent times. As highlighted by Moulin-Stożek et al. [19], “in the UK, recourse to virtue has had a pivotal role in the last decade in politics, as lauded in the “Red Tory” or “Blue Labour” movements, for example”. In essence, the author argued that the desired educational outcomes of the near future will not be confined to the traditional priorities of literacy, numeracy, and scientific knowledge alone. Instead, dispositions, values, or personal attributes are increasingly more important in the attempt to secure economic prosperity and social stability.

This approach is in line with the recent framework on value-based healthcare proposed by the Expert Panel on Effective Ways of Investing in Health (EXPH) of the European Commission [21]. This new framework is built on four value pillars: appropriate care to achieve patients’ personal goals (personal value), achievement of best possible outcomes with available resources (technical value), equitable resource distribution across all patient groups (allocative value), and contribution of healthcare to social participation and connectedness (societal value). The values framework proposed by EXPH has also been applied to the vaccination field [22,23], emphasizing that personal and social values are not yet much investigated and taken into consideration. Instead, the understanding of the broad value of vaccination and the effective translation of this knowledge to the different stakeholders is essential to strengthen vaccination policies and strategies and counteract disinformation and misinformation [24].

In addition, it noteworthy that attempts to reconcile virtue ethics with traditional interpretations of social justice have been made philosophically; L. Tessman’s [25] notion of feminist “critical virtue ethics” is an example of this. Another interesting example is the personalistic approach to bioethics proposed by E. Sgreccia [26,27,28,29,30], which brought together solidarity with the attention to the individual (common good is interpreted as a sum of individual goods which require positive attitudes).

Finally, the suggestion that virtues are pertinent to public health practice is not a new concept [31,32,33,34]; there is a growing body of works on this, most of which are specifically dedicated to the pandemic [19,35,36]. Among them, Fowers et al. [36] is perhaps the most systematic conceptual framework for virtue as a response to the pandemic. More importantly, it focuses on how three specific virtues can address the difficulties of risk, injustice, and complexity exacerbated by the pandemic. Our article is in line with this emerging perspective.

Our analysis highlights important lessons: ethics behind public health interventions should be made clearly explicit and should be discussed and perspective of the agent should receive greater consideration. In turn, this poses the question on how to acquire the perspective of the agent.

## 5. Limitations

The present analysis has two important limitations. In the first place, it refers only to the Italian context, while a broader perspective would be necessary. For example, it would be useful to carry out a comparative work with other countries. In the second place, the arguments provided to support the main theses should be corroborated by social scientific studies. For instance, the thesis that, in Italy, the debate on criteria for prioritizing the anti-COVID-19 vaccine was insufficient reflects more the authors’ impression than the result of scientific studies.

## 6. Conclusions

Two lessons can be drawn from our analysis: first, public health measures are not primarily a “scientific matter” but, rather, a political and value-laden decision. Hence, implementation of (pandemic) vaccination programs, as well as any kind of public health measure, should clearly be based on a combination of ethical reasoning and technical-scientific data as well as a values-based approach. Secondly, neither deontologism nor consequentialism are capable of answering the question “what should I do?” and this can lead to “personal alienation”. Integrating or at least recognizing the importance of appealing to virtues and/or the perspective of the agent can be an added value in order to improve public health measures. In recent times, attempts to reconcile virtue ethics with traditional interpretations of social justice are arising and the present article is in line with this perspective. In turn, this poses the question on how to acquire the perspective of the agent.

Starting with the realization that there may be different views in society, it becomes fundamental to learn how to explore and address this pluralism of values. There is no easy way to carry out this task. One solution may come from “scoping” exercises, for example, conducting interviews with citizens and participatory observations (qualitative research) or a combination of the two [37]. This does not mean that scoping should be necessarily directed to find out who (or what) is right and who (or what) is wrong; rather, it may help participants to come to realize the complex nature of a certain issue in the first place. Moreover, scoping exercises may help to understand the different perspectives in society with respect to the desirability of a certain aim, as well as to understand how society interprets certain moral values. On this line, public health interventions should be more the result of a process that includes and integrates the perspectives of different agents, rather than the emanation of technical-scientific organizations. Therefore, integrating scoping exercises may be one possible direction to take in the future. However, in our opinion, this may be easier said than done, since it will be challenging to find a common ground.

## Data Availability

No new data were created or analyzed in this study. Data sharing is not applicable to this article.

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
