# Peer review of "What Type of Person Should I Be? About the Appeal to Virtues in Public Health Interventions"

_vaccines, 2023, doi:10.3390/vaccines11040767_

Round 1

Reviewer 1 Report

Review: What type of person should I be? About the appeal to virtues in public health interventions.

This is an interesting and important submission highlighting the political and ethical nature of public health measures in relation to the COVID-19 pandemic, as well as the need to integrate or at least recognize the value of the appeal to virtues in such public health measures. The authors demonstrate a clear and practical understanding of the notions of deontological, consequentialist and virtues approaches to ethics generally and in relation to public health measures, all of which is especially pertinent to the ways in which such measure have/have not been applied during the pandemic, and how public health measures are primarily not scientific matters, but rather political and value-laden decisions.

However, I have some questions in relation to the methodology, and how (or indeed whether) the results actually emanate from the critical interpretive review that was conducted. This is not clear. While I can see how 3.1-Ethics behind public health interventions and 3.5-Italian COVID-19 vaccination program, may have come out of the critical interpretive review, when I read the results more broadly, it seems to me that sections 3.2, 3.3 and 3.4 – Deontological approach, Utilitarian approach, and Virtue ethics respectively, are not actually findings or results, but are rather ethical perspectives that ought rightly to be discussed, but which should be situated in the framing section of the submission (in the introduction). The results could then contain a section that shows the degree to which these approaches are, or are not, reflected in the documents that formed part of the critical interpretive review. It may be that I have misunderstood this, but if it is not clear to me, then it may well not be clear to the readership of this journal, and hence I recommend that the authors revisit this section of their submission.

I relation to the materials and methods, I would like some further information. Can you detail further exactly what is involved in a critical interpretive review to demonstrate its rigour? What was the total number of papers identified through the critical interpretive review process, and what were the inclusion and exclusion criteria that resulted in 10 papers that were included in the review? Does that fact that there were only 10 papers included require some justification? One would ordinarily expect a greater number of published papers to be included in the final selection in such a review.

The conclusion also requires a little further elaboration. For example, what are the limitations of the research? Are there any avenues for further or future research that could be recommended and explored? What are the implications of your study for policy concerning the direction and implementation of public health measures with regard to ethical concerns?

I encourage the authors to revise their paper in line with my comments, as I do think such a revision would make a valuable contribution to this journal.

Author Response

Dear Reviewer 1,

we appreciate the time and effort that you have dedicated to providing your valuable feedback on our manuscript. We are grateful for your suggestions and comments on our paper. We were able to incorporate changes to reflect all of the suggestions provided. We have highlighted such changes within the manuscript.

Here is a point-by-point response to the comments and concerns.

Comments about methodology

Yes, you are absolutely right when you say that methodology is not satisfactory. Part of the problem derives from the fact that it is not easy to explicit methodology in this kind of article that are quite free. In fact, we initially chose to indicate the critical interpretive review (a methodology that we know and that we applied in other articles), since it seemed to us the closest to the “path” we followed. However, we are aware that it was not exactly the same and that the final result could raise questions. Therefore, we decided to remove that reference and to replace it with a clear description of the work done. We hope the new version is useful to better understand the methods used.

Comments about limitations

We inserted the main limitations of the research. Thank you for having raise the question.

Comments about future research

We included a section dedicated to possible future developments of our reasearch. Thank you for your suggestion. 

Reviewer 2 Report

The article entitled:

What type of person should I be? About the appeal to virtues in 2 public health interventions

It is an interesting article, important in our field.

However, my suggestions for improvement are:

Improve the discussion:

Broaden debate with more discussion and critical reflection.

Incorporate theoretical implications

Incorporate practical implications

Incorporate limitations

Incorporate future lines of research

In the final conclusions, indicate better or clearly what you have found out with this and what would be necessary for future interventions to take into account, because otherwise, it would be more of the same in an area where there is everything, but nothing clear.

I hope you get better visibility this way! Congratulations, I loved your work!

My sincere congratulations for the work.

Author Response

Dear Reviewer 2,

we appreciate the time and effort that you have dedicated to providing your valuable feedback on our manuscript. We are grateful for your suggestions and comments on our paper. We were able to incorporate changes to reflect almost all of the suggestions provided. We have highlighted such changes within the manuscript.

Here is a point-by-point response to the comments and concerns.

Comments about discussion, debate with more discussion and critical reflection, theoretical implications, and practical implications

Thank you for your suggestions. We improved the discussion by clearly highlighting, from a practical point of view, the main results of our investigation.  

Comments about limitations

We inserted the main limitations of the research. Thank you for having raise the question.

Comments about future research

We included a section dedicated to possible future developments of our reasearch, taking into consideration your suggestion to be practical. Thank you for your suggestion.